# Dendritic Spine in Autism Genetics: Whole-Exome Sequencing Identifying De Novo Variant of *CTTNBP2* in a Quad Family Affected by Autism Spectrum Disorder

**DOI:** 10.3390/children10010080

**Published:** 2022-12-30

**Authors:** Yingmei Xie, Hui Wang, Bing Hu, Xueli Zhang, Aiping Liu, Chunquan Cai, Shijun Li, Cheng Chen, Zhangxing Wang, Zhaoqing Yin, Mingbang Wang

**Affiliations:** 1Division of Neonatology, Longgang District Maternal and Child Health Hospital, Shenzhen 518172, China; 2Division of Child Health Care, Xiamen Branch of Children’s Hospital of Fudan University (Xiamen Children’s Hospital), Xiamen 361006, China; 3Division of Neonatology, Shenzhen Longhua People’s Hospital, Shenzhen 518109, China; 4The Department of Laboratory, Baoan Public Health Service Center of Shenzhen, Shenzhen 518108, China; 5Tianjin Key Laboratory of Birth Defects for Prevention and Treatment, Tianjin Children’s Hospital (Children’s Hospital of Tianjin University), Tianjin Pediatric Research Institute, Tianjin 300134, China; 6First Medical Center, Chinese PLA General Hospital, Department of Radiology, Beijing 100853, China; 7The People’s Hospital of Dehong Autonomous Prefecture, Division of Pediatrics, Dehong Hospital of Kun-ming Medical University, Mangshi 678400, China; 8Microbiome Therapy Center, South China Hospital, Medical School, Shenzhen University, Shenzhen 518116, China; 9Shanghai Key Laboratory of Birth Defects, Division of Neonatology, Children’s Hospital of Fudan University, National Center for Children’s Health, Shanghai 201102, China

**Keywords:** autism spectrum disorder, whole-exome sequencing, *CTTNBP2*, cortactin-binding protein 2

## Abstract

Autism spectrum disorder (ASD) affects around 1% of children with no effective blood test or cure. Recent studies have suggested that these are neurological disorders with a strong genetic basis and that they are associated with the abnormal formation of dendritic spines. Chromosome microarray (CMA) together with high-throughput sequencing technology has been used as a powerful tool to identify new candidate genes for ASD. In the present study, CMA was first used to scan for genome-wide copy number variants in a proband, and no clinically significant copy number variants were found. Whole-exome sequencing (WES) was used further for genetic testing of the whole quad family affected by ASD, including the proband, his non-autistic sister, and his parents. Sanger sequencing and MassARRAY-based validation were used to identify and confirm variants associated with ASD. WES yielded a 151-fold coverage depth for each sample. A total of 98.65% of the targeted whole-exome region was covered at >20-fold depth. A de novo variant in *CTTNBP2*, p.M115T, was identified. The *CTTNBP2* gene belongs to a family of ankyrin repeat domain-containing proteins associated with dendritic spine formation. Although *CTTNBP2* has been associated with ASD, limited studies have been developed to identify clinically relevant de novo mutations of *CTTNBP2* in children with ASD; family-based WES successfully identified a clinically relevant mutation in the *CTTNBP2* gene in a quad family affected by ASD. Considering the neuron-specific expression of *CTTNBP2* and its role in dendritic spine formation, our results suggest a correlation between the *CTTNBP2* mutation and ASD, providing genetic evidence for ASD spine pathology. Although the present study is currently insufficient to support the assertion that the de novo mutation M115T in *CTTNBP2* directly causes the autism phenotype, our study provides support for the assertion that this mutation is a candidate clinically relevant variant in autism.

## 1. Introduction

Autism spectrum disorder (ASD), a group of lifelong neurodevelopmental disorders, affects more than 78 million people worldwide and identifying the etiology of autism and developing targeted pharmacological treatment options is a priority [1,2]. Genetic background is thought to play an increasingly important role in the development of autism, and the evidence supporting this assertion comes primarily from studies of twins, which show that identical twins are significantly more likely to share autism than dizygotic twins; a 50–80% heritability of autism has been found [3,4,5,6]. The prevalence of autism is gradually increasing, and the arguments in support of this fact come primarily from epidemiological survey data from the Centers for Disease Control and Prevention (CDC) in the US. The CDC data indicate that the prevalence of ASD has increased from an estimated 1 in 166 in 2004 to 1 in 44 in 2018, or almost tripled [7]. This observation also challenges the classical hypothesis of the genetic etiology of ASD. Notably, the contribution of environmental factors to ASD has also been increasingly noted [6]. As opposed to genetic factors, the role of environmental factors in ASD appears to be more complex. New studies suggest that environmental factors may play a role in autism development by interacting with genes. For example, environmental factors may directly influence the production of de novo mutations or affect DNA methylation [8,9].

The widespread use of large-scale high-throughput sequencing in autism research [10,11,12,13,14] has directly led to the discovery of a large number of autism-causing genes or related genes, and, as of the date of writing of the manuscript (1 November 2022), as many as 1095 autism-causing genes or related genes were included in the SFARI Gene database (https://gene.sfari.org/database/gene-scoring/ (accessed on 1 November 2022)). Notably, in addition to rare variants inherited from parents, germline de novo variants and chimeric mutations contribute significantly to autism [15]. However, the discovery of a large number of autism genes has been accompanied by the identification of a few genes with high-frequency mutations in autistic families, which has made it difficult to carry out mechanistic studies in genetically modified animal models of autism [16]; encouragingly, patient-derived induced pluripotent stem cells or organoid-based studies are beginning to reveal commonalities in the etiology of autism on different genetic mutation backgrounds [17]. In the meantime, transcriptional-level studies of autism and biological pathway enrichment analyses have revealed a number of genes that control synapse formation and signaling as well as genes that control transcription and chromatin conformation to regulate gene expression and play a key role in autism [18,19,20,21].

Genes that control synapse formation and synaptic signaling activity are an important class of ASD-causing or related genes, and, among these synapse-associated ASD candidates, human genetic evidence has shown that cortactin-binding protein 2 (*CTTNBP2*) is a strong ASD candidate gene [22,23,24,25]. Gene modification studies at the cellular level and in animal models have linked ASD-associated mutations in *CTTNBP2* to altered neuronal function and the production of autism-like behaviors [26]. The excitatory synapses in the mammalian nervous system are located in dendritic spines, which are the neuron-specific subcellular structures necessary for neural circuitry and function [27,28]. Recent evidence suggests that an alteration in dendritic spines may be involved in the pathogenesis of ASD [29,30].

Despite the complex genetic background of ASD, genetic research has largely advanced our understanding of the causes of ASD and identified the molecular pathways involved in the pathology. Whole-genome/exome sequencing, candidate gene resequencing, and copy number variation analysis have identified many ASD-related variants [31]. Currently, ASD can be described as a genetic model of a “common disease, many rare mutations” [32]. In the context of ASD-associated mutations exhibiting rare or even "private" but high penetrance; the genetic model also highlights the importance of identifying the functional pathways associated with ASD risk genes. This situation, on the one hand, may facilitate the discovery of new candidate genes for ASD, while, on the other hand, it may serve as a target for therapeutic drug intervention. Genetic studies targeting ASD have identified a large number of autism risk genes. To date, the SFARI Gene database includes more than 1000 ASD candidate genes. Among these genes, a subset of genes that are key regulators of synaptic plasticity has attracted increasing attention. These genes include those encoding synaptic proteins, small GTPases, cell adhesion molecules, and actin cytoskeletal proteins. The functional variants of these genes, identified in ASD families, have some clinical significance. This is mainly because the functional mutations in these genes may alter the synaptic strength or number, ultimately altering neuronal connectivity in the brain and increasing an individual’s risk of developing ASD. At the same time, they also hold promise for the development of targeted pharmacological intervention programs [15].

## 2. Materials and Methods

### 2.1. Participants

The proband was a 5-year-old boy who had been diagnosed with ASD in accordance with the Diagnostic and Statistical Manual of Mental Disorders, Fifth Edition (DSM-5), at 1 year and 6 months old. He was delivered at term by Cesarean section when his mother was 31 years old. There were no abnormalities at birth and no clear family history of neurological disorders. The proband could only say monosyllabic words, could not construct sentences, communicated little, often ignored people, only glanced at others when dealing with someone, did not actively participate in activities with other children, and liked rotating objects and mobile phones, as well as closing doors and switching things on and off. His Autistic Behavior Checklist score was 24, and his Childhood Autism Rating Scale score was 33. Brain magnetic resonance imaging examination showed no abnormal parenchymal brain. Emission computed tomography (ECT) examination showed focal perfusion and dysfunction (Appendix A). We obtained informed consent from the parents of the child. This study was conducted in accordance with the Declaration of Helsinki and was approved by the Ethics Committee of the Children’s Hospital of Fudan University. Three to five milliliters of whole blood was drawn from all four members of the quad family. The Magbind Blood DNA Kit (CW Biotech, Beijing, China) was used to exact the genomic DNA in accordance with the manufacturer’s instructions, and the exacted DNA was stored in an ultra-low temperature refrigerator at −80 °C before use.

### 2.2. Chromosome Microarray

Chromosome microarrays were performed as described in previous reports [33,34]. Briefly, first, the Affymetrix CytoScanHD platform (Thermo Fisher, Hudson, NH, USA) was used for genotyping and completed according to the manufacturer’s instructions. Then, the ChAS software package (Thermo Fisher, Hudson, NH, USA) was used to identify copy number variants (CNVs); the cutoffs used are 200 kb for gain, 100 kb for loss, and 10,000 kb for the region of homozygosity (ROH). Then, the determined CNVs were annotated in accordance with the Database of Genomic Variants (DGV; http://projects.tcag.ca/variation/ (accessed on 26 November 2022)), after which the UCSC Browser based on GRCH37 was used to determine the gene composition of CNVs of interest. Finally, each of the CNVs was further evaluated using an in-house benign CNV database. At the same time, we analyzed the possibility that the affected genes were associated with the phenotypic characteristics of ASD observed in the patients.

### 2.3. Whole-Exome Sequencing

Based on the previously published study in [34], we completed whole-exome sequencing of the quad family. In brief, a Qubit dsDNA BR assay kit (Thermo Fisher, Hudson, NH, USA) was used to determine the DNA concentrations; then, the TruSeq DNA Sample Preparation kit (Illumina, San Diego, CA, USA) was used to construct 200–500-bp libraries, and 0.5 micrograms genomic DNA was used. Subsequently, SureSelect Human All-Exon V5 (Agilent, Santa Clara, CA, USA) was used to capture the targeted region in the libraries. Finally, the captured libraries were enriched, and Illumina Hiseq X-ten (Illumina, San Diego, CA, USA) was used to sequence the captured libraries.

### 2.4. Bioinformatic Analysis

The de novo and inherited variants of the proband were called a pipeline per our previous studies [11]. In brief, the reads with adaptor sequences or with low quality were first filtered out using Trimmomatic [35]. Subsequently, the Burrows–Wheeler Aligner [36] was used to align the remaining reads to the reference human genome (hg19), after which Picard (http://broadinstitute.github.io/picard/ (accessed on 25 August 2019)) was used to remove polymerase chain reaction (PCR) duplicates; then, Genome Analysis Toolkit [37] version 3.0 was used to call variants in each sample, followed by the use of mirTrios [38] to identify the de novo or rare inherited variants in the proband. Finally, ANNOVAR [39] was used to annotate the variants. To prioritize variants, we first filtered out the common variants (with a minor allele frequency of >0.05 in the 1000 Genomes Project database, gnomAD database, or our in-house database); then, we focused on the variants potentially associated with ASD according to the Simons Foundation Autism Research Initiative (SFARI) gene database [40] or predicted by Phenolyzer [41]. Finally, we focused on variants that might damage protein function as predicted by SIFT [42], PolyPhen-2 [43], and MutationTaster [44].

### 2.5. Sanger Sequencing

To confirm the candidate variant *CTTNBP2,* NM_033427, exon 3, c.344T > C, p.M115T, the ABI9700 PCR system (Thermo Fisher, Hudson, NH, USA) was used to perform a PCR using primers (forward, 5′-GTTGTGGGGAAAATGACCAT-3′; reverse, 5′-ATGACCCGTTCCTGGCACTC-3′). The ABI 3730xl sequencer (Thermo Fisher, Hudson, NH, USA) was used to sequence the obtained PCR product.

### 2.6. Genotyping

A total of 867 individuals not affected by ASD were recruited for validation of the variant frequency. The Magbind Blood DNA Kit (CW Biotech, Beijing, China) was used to extract the genomic DNA. *CTTNBP2* p.M115T (c.344T > C) was further confirmed using the MassARRAY iPLEX platform (Agena Bioscience, San Diego, CA, USA), in accordance with the manufacturer’s manual [45].

### 2.7. Tertiary Structure Prediction

We used alphafold2 [46,47] to predict the tertiary structure of wild-type and mutant gene proteins based on their amino acid sequences. The maximum template date is 14 May 2022, and the AlphaFold model is a monomer.

### 2.8. Genetic Model Prediction

The prediction of the genetic model was done by DOMINO software [48]. The DOMINO score obtained was taken between 0 and 1, suggesting the probability of conforming to the dominant genetic model. The closer the score is to 0, the more likely it is the recessive genetic model; the closer the score is to 1, the more likely it is a dominant genetic model.

### 2.9. Genes Enrichment Analysis

The common variants, with an allele frequency greater than 0.05 in public databases, and synonymous mutations were filtered out first. Then, all variant genes were used to perform gene enrichment analysis. Finally, an enrichment analysis of all variant genes was performed by Metascape software [49], and the terms enriched included gene ontology (GO), pathway terms from the Kyoto Encyclopedia of Genes and Genomes (KEGG), canonical pathways, hallmark gene sets, etc.

## 3. Results

### Variants Identified of the Proband

In order to find the genetic mutations associated with autism, we first examined the genome-wide copy number variation (CNV) by chromosome microarray (CMA) technology. The proband affected with ASD was first screened by CMA, which resulted in no clinically relevant CNVs being identified (Appendix A). Then, the proband, together with his sister and parents who were not affected by ASD, were subjected to WES (Appendix A). WES yielded 12.12 Gbp, or 151-fold depth per sample; 98.65% of the targeted whole-exome region was covered at >20-fold depth, and the overall coverage rate was 99.89% per sample (Appendix A). On average, 22,218 exonic single-nucleotide variants (SNVs) (Appendix A) and 690 exonic small insertions and deletions (indels) (Appendix A) were called for each sample.

After initially filtering the public database (1000 genome and genome) for common variants and synonymous variants, a total of 541 variants were obtained (see Appendix A) from 248 genes in order to better perform gene enrichment analysis of non-SFARI genes and discover some biological pathways associated with the candidate autism genes. After removing the SFARI genes (Appendix A, total 1095 genes), a total of 218 non-SFARI genes were obtained. We performed a pathway enrichment analysis for both 248 genes and 218 non-SFARI genes using Metascape; the results are shown in Appendix A. It can be seen that all 248 genes were enriched to delayed speech and language development and Byzantine-arch palate pathways; the 218 non-SFARI genes were enriched to hyperinsulinism and Byzantine-arch palate pathways.

Considering that we have whole-exome sequencing data for the complete quad family, we can easily distinguish the variants found in the proband into inherited or de novo variants, and all of the variants in the proband were used together to find the clinically relevant variants in autism. Through systematic bioinformatic analysis (Figure 1), we identified candidate variants in some known ASD-related genes [40]. Combining this with an analysis using a phenotype-based prioritization tool [41], p.M115T (chr7: 117450889, *CTTNBP2*, NM_033427, exon 3, c.344T > C), a de novo missense variant in *CTTNBP2*, was shown to be associated with ASD (Figure 2a, Appendix A). This variant was further confirmed by Sanger sequencing (Figure 2b).

To determine the allele frequency of nonsynonymous mutations of the *CTTNBP2* gene in the general population, we downloaded the *CTTNBP2* gene variants included in the gnomAD database (see Appendix A) and performed the statistics. We found that the allele frequency of missense mutations in the *CTTNBP2* gene ranged from 0 to 0.0019 in the East Asian population, which is relatively rare; notably, we found that the allele frequency of missense mutations in the *CTTNBP2* gene located in the coiled-coil (CC) region ranged from 0.000283 to 0.00000397, which is very rare. Notably, we found the presence of variant p.Met115Lys in the *CTTNBP2* gene, which is rare in the population (only 1 detected in 31,392 individuals) and rare in the East Asian population (0 detected in 1560 East Asians).

Considering that the collection of a Chinese population may be limited to the East Asian population of the gnomAD database, we performed further large-scale validation of M115T in the Chinese population. The MassARRAY confirmed that the variant was rare (0 in 867) in Chinese individuals not affected by ASD (Appendix A).

The *CTTNBP2* (cortactin-binding protein 2) gene belongs to a family of ankyrin repeat domain-containing proteins that function in protein–protein interactions; the missense variant p.M115T leads to a change from methionine to threonine at amino acid sequence position 115; the missense variant p.M115T was predicted by SIFT [42], PolyPhen2 [43], and MutationTaster [44] to damage the function of the *CTTNBP2* protein. The variant p.M115T is located in the N-terminal domain of CTTNBP2. *CTTNBP2* was also identified as an ASD-associated gene and classified as having “suggestive evidence” of this by the SFARI Gene database [40].

## 4. Discussion

ASD affects around 1% of children with no effective blood test or cure [50]. Studies have shown that genetic mechanisms play an important role in the pathogenesis of ASD [51,52]. To date, over 1000 autism-related genes have been included in the SFARI database. These genes help to understand the etiology of autism occurrence and identify potential pharmacological interventions. Genetic mutations of these genes found in autism-affected families are an important reference for the diagnosis of autism. Currently, chromosome microarray, together with high-throughput sequencing has been used for clinical genetic diagnosis of ASD [53]. We have accumulated some experience in discovering autism-associated variants by high-throughput sequencing of trio or quad families affected by autism. With 32 ASD Canadian families with negative CMA results, we identified clinically relevant variants in 16 of 32 families and developed a pipeline to identify rare de novo and inherited variants relevant to ASD [11]. In the present study, a quad family affected by ASD was recruited. The proband was first screened by CMA with negative results, after which WES was performed on the proband, together with his sister and his parents who were not affected by ASD. Through systematic bioinformatic analysis, a de novo mutation of *CTTNBP2*, p.M115T, was successfully identified and confirmed by Sanger sequencing as the clinically relevant variant in the proband. This shows that family-based WES is a viable method for genetic testing for ASD.

The *CTTNBP2* gene is located on chromosome 7q31, a candidate region for autism [54]. The *CTTNBP2* gene encodes a neuron-specific cytoskeleton protein, which controls the formation and maintenance of dendritic spines by interacting with cortactin [55,56]. Cortactin is a ubiquitous protein that regulates the branching and stability of F-actin. Although cortactin is not neuron-specific, it is widely distributed in the dendritic spine area and is thought to regulate the remodeling of neuronal activity. Based on the analysis of the secondary structure of an amino acid sequence, the CTTNBP2 protein is composed of an N-terminal coiled-coil domain and a C-terminal intrinsically disordered region (IDR) [26]. The coiled-coil domain mediates the oligomerization of CTTNBP2 as well as both homopolymerization and heteropolymerization with the striatum [56]. The IDR region at the C-terminal consists of two regions, namely an intermediate domain responsible for microtubule interaction and a proline-rich region and a region of about 100 amino acids that mediates the interaction with cortactin (Figure 3a, refer to [26]). The M115T mutation found in the coiled-coil domain is noteworthy because Shih et al. found that the M120I mutation is also located in the coiled-coil nearby and has a negative impact on the interaction between CTTNBP2 and cortactin. After introducing the patient’s *CTTNBP2* gene with the heterozygous mutation M120I into mice, the mice exhibited reduced social interactions [26]. Considering that M115T and M120I are more proximate (Figure 3b), it is not difficult to speculate that M115T may also play a very important role in ASD.

So far, 38 nonsynonymous mutations in the *CTTNBP2* gene have been identified in ASD patients (Appendix A), and most of these mutations are located in the proline-rich region and the C-terminus (Figure 3c). It is known that the C-terminus mediates the binding of the CTTNBP2 protein to the cortactin protein and may have a more direct effect on CTTNBP2 function than other regions. Shih et al. found that mutations in R533*, located in the proline-rich region, directly led to a reduction in the number of synaptic spines [26,57]. A recent study revealed that zinc binds to the N-terminal coiled-coil region of CTTNBP2 and promotes higher-order assembly, which can rescue the autistic phenotype in mice [58]. The *CTTNBP2* gene is a known strong autism candidate gene in the SFARI database. We identified a rare de novo missense mutation, p.M115T, in the *CTTNBP2* gene in patients affected by autism. This mutation is of interest because of its location in the N-terminal coiled-coil (CC) region, which mediates the oligomerization of the CTTNBP2 protein and thus regulates spine formation. The next step, which is in-depth functional validation, is required to confirm whether this variant directly affects spine formation and the causal relationship with autism.

Considering the complexity of the genetic model of autism, the currently identified autism genes sometimes conform to an autosomal recessive genetic model, sometimes to an autosomal dominant genetic model, and sometimes are difficult to explain by the conventional genetic model. No other mutations have been found in the *CTTNBP2* gene, except for the de novo heterozygous mutation in p.M115T. Moreover, studies in animal models have shown that the introduction of heterozygous mutations into the *CTTNBP2* gene can lead to behavioral defects in mice. These suggest that the probability of being consistent with autosomal dominant is greater than autosomal recessive. The genetic model of autism is complex, with the emerging double-hit, or multiple-hit hypotheses [59], that is, mutations in two or more genes together cause autism. In particular, this hypothesis has also been used by some scholars when interpreting the detection of de novo mutations in autistic patients [60]. Finally, based on the results of DOMINO, the possibility that p.M115T de novo heterozygous mutations is consistent with the double-hit or multiple-hit hypotheses cannot be ruled out either. Also there is the possibility that the variant could be functioning molecularly as recessive while functioning genetically as dominant. For example, if the detected variant is causing excessive polymerization, of CTTNBP2, there may be little oligomeric or monomeric CTTNBP2 available to interact with other cellular interactors, resulting in insufficient CTTNBP2 functional activity. This is molecularly recessive, consistent with the results of the DOMINO analysis, but also would be genetically dominant.

Highly specialized neuronal morphology provides a substrate for constructing human cognitive and behavioral neural circuits, and neuron morphogenesis is a very complex process, which is affected by both genetic susceptibility and environmental events [61]. In the mammalian nervous system, dendritic spines are located at excitatory synaptic sites; they are neuron-specific subcellular structures that are necessary for neural circuitry and function. Hundreds of genes are known to be involved in the regulation of dendritic spine formation; most of these genes are non-neuron specific, including *ANXA1*, syndecan-2, *NF1* (encoding neurofibromin), *VCP* (valosin-containing protein), *CASK*, and *CTTNBP2* [29,62]. *CTTNBP2* is a neuron-specific gene that regulates dendritic spine formation [29]. *CTTNBP2* is located in a region associated with ASD, namely, the autism candidate region at 7q31 [54]. Iossifov et al. first identified a de novo 2-bp frameshift deletion at this site in patients affected with ASD [23], which suggested that *CTTNBP2* is an ASD-associated gene. In agreement with this, we found a de novo mutation of *CTTNBP2*, p.M115T. This mutation, located at the N-terminus-binding domain, leads to a change from methionine to threonine at amino acid position 115 and may affect protein–protein interactions. *CTTNBP2* plays an important role in regulating dendritic spine morphology and synaptic plasticity [55] in synapse formation. Furthermore, MacGillavry et al. found that cortactin can interact with Shank to control actin dynamics and maintain the flexibility of neuronal spines and synapses. A mutation of *CTTNBP2* that might damage the function of the protein it encodes could thus potentially lead to abnormal brain connectivity [63]. In agreement with this, an ECT image of the proband showed abnormal focal perfusion. The present study suggests that family WES is a viable approach for identifying the genetic mutations associated with ASD. The *CTTNBP2* mutation that we found in this study may well explain the patient’s phenotype.

## 5. Conclusions

We successfully identified a clinically relevant de novo mutation in the *CTTNBP2* gene in a quad family affected by ASD. Considering the neuron-specific expression of *CTTNBP2* and its role in neuronal and dendritic spine formation, this finding provides genetic evidence for the spine pathology of ASD. However, this study has some limitations: First, we do not know whether this de novo mutation of the *CTTNBP2* gene is common in ASD patients or how this mutation participates in the occurrence of ASD, so screening for the *CTTNBP2* mutation in large-scale ASD patients and in-depth functional studies will be necessary to determine the role of *CTTNBP2* in the pathogenesis of ASD. Moreover, common genetic variants do play an important role in autism, and, considering the limited sample of this study, we did not focus on common genetic variants. In conclusion, although the present study is currently insufficient to confirm that the de novo mutation M115T in *CTTNBP2* directly causes the autism phenotype, our study provides support for the assertion that *CTTNBP2* is a strong candidate gene in autism.

## Figures and Tables

**Figure 1 children-10-00080-f001:**
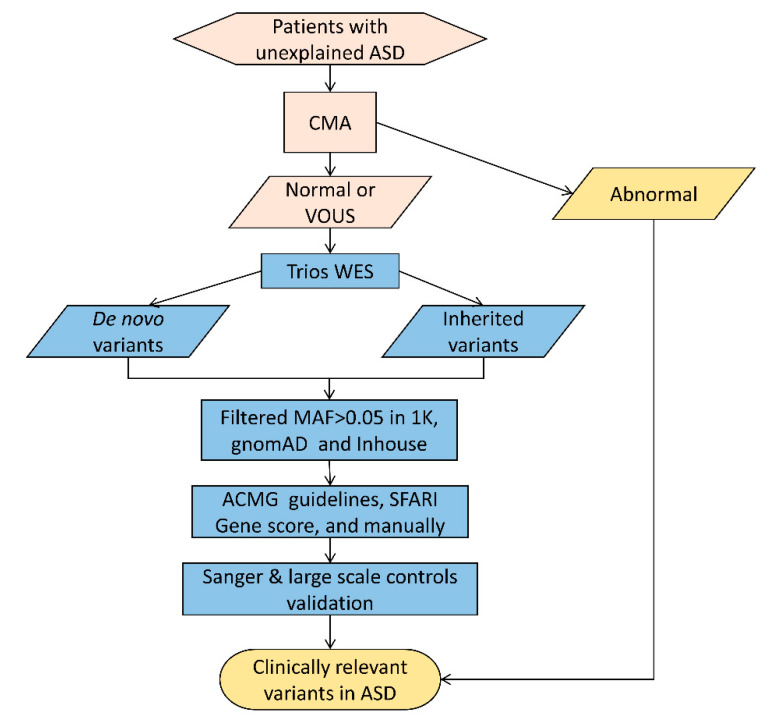
Flowchart for detecting clinically relevant variants in ASD by CMA and WES. ASD, autism spectrum disorder; CMA, chromosome microarray; HPO, human phenotype ontology; MAF, minor allele frequency; SFARI, Simons Foundation Autism Research Initiative; VOUS, variant of uncertain clinical significance; WES, whole-exome sequencing. First, we filtered for common variants with an allele frequency greater than 0.05; then, we evaluated the pathogenicity of the variants with reference to the ACMG guidelines, together with the SFARI gene score to identify candidate clinically relevant variants for autism. Meanwhile, considering the limited data on the Chinese population within the East Asian data in gnomAD databases, we validated clinically relevant variants that could be candidates for autism in 867 normal populations.

**Figure 2 children-10-00080-f002:**
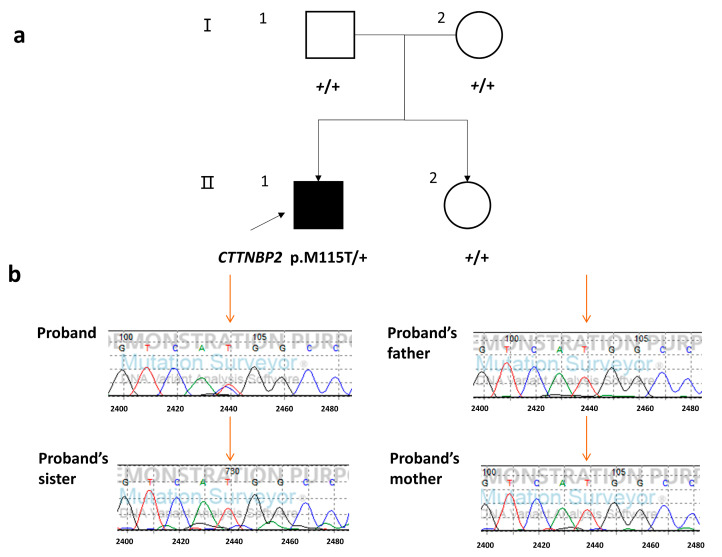
De novo *CTTNBP2* variant confirmed in a quad family affected by ASD: (**a**) *CTTNBP2* p.M115T (c.344T > C) was identified in the proband as a de novo mutation (II-1), +, wild type; (**b**) Sanger sequencing confirmed the presence of *CTTNBP2* p.M115T (c.344T > C) in the proband as a de novo mutation.

**Figure 3 children-10-00080-f003:**
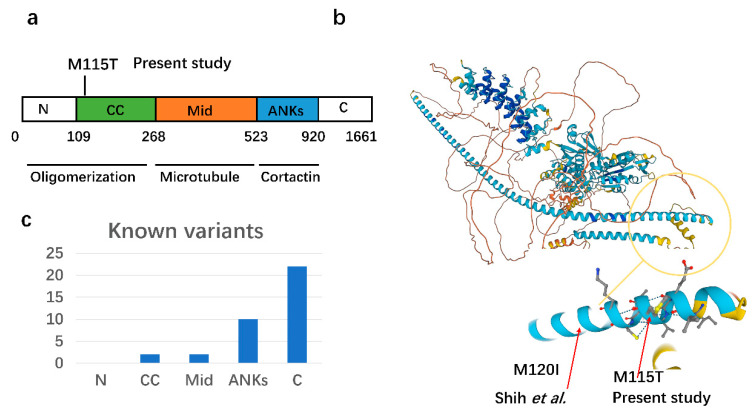
Structure of the M115T mutation in the CTTNBP2 encoded protein and known CTTNBP2 mutation statistics. (**a**) Illustration of the important structural domain of the protein encoded by the *CTTNBP2* gene. The M115T mutation is located in the N-terminal coiled-coil (CC) region, which mediates the oligomerization of the CTTNBP2 protein and can affect the CTTNBP2 protein binding to the cortactin protein. Shih et al. found that the M120I mutation in the N-terminal region of CTTNBP2 protein negatively affected the interaction of CTTNBP2 with cortactin; mice exhibited reduced social interaction after introducing the patient’s *CTTNBP2* gene heterozygous mutation M120I into mice [26]. The M115T mutation identified in the present study is very close to M120I. (**b**) A 3D structural illustration of the protein encoding CTTNBP2, obtained by AlphaFold [46,47] Protein Structure Database (https://alphafold.com/entry/Q8WZ74, accessed on 1 November 2022). (**c**) Bar plot of ASD risk variants identified in the *CTTNBP2* gene according to different regions featuring 39 variants in SFARI gene database (https://gene.sfari.org/database/human-gene/CTTNBP2#variants-tab (accessed on 1 November 2022)); one of the synonymous mutations was removed.

## Data Availability

The data used in this study have been uploaded to the supplementary annex. Considering the control of human genetic resources, the original data have not been uploaded to the public database, and the VCF file for research can be obtained from the corresponding author.

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
