# Peer review of "Dendritic Spine in Autism Genetics: Whole-Exome Sequencing Identifying De Novo Variant of *CTTNBP2* in a Quad Family Affected by Autism Spectrum Disorder"

_children, 2022, doi:10.3390/children10010080_

Round 1

Reviewer 1 Report

Yingmei Xie et al. found a de novo missense variant in CTTNBP2 in an inidividual with ASD.

The experimental work and bioinformatic analysis are correct and following standards. However, the interpretation is in my opinion incorrect.

There is not sufficient evidence to link this variant to ASD.:

- the gene PLi is 0 and has many missenses in gnomAD

- The DOMINO score is of 0.096 (probability to be associated with a dominant disorder)

- There is 39 variants in SFARI database but only 3 of them are de novo and only one of them is a missense variant.

- Finding a de novo variant does not imply pathogenicity. Every individual carry 1 to 3 coding de novo variant and most of them are benign. According to ACMG guidelines, the variant can only be classified as VUS (PS2, PM2, PP3).

- The variant M120I identified by Shih et al. was not found de novo but maternally transmitted (in SFARI).

In conclusion without either experimental data proving a gain-of-function mechanism of the missense variant or other cases with missense de novo variant in this gene, there is not enough evidence to link the variant to the phenotype. Otherwise, we could sequence any individual, find de novo variant with a link to a disease and claim that there is an causal effect.

Author Response

Comments and Suggestions for Authors

Yingmei Xie et al. found a de novo missense variant in CTTNBP2 in an inidividual with ASD.

The experimental work and bioinformatic analysis are correct and following standards. However, the interpretation is in my opinion incorrect.

There is not sufficient evidence to link this variant to ASD.:

- the gene PLi is 0 and has many missenses in gnomAD

Reply: Thanks for the comment, we know that Probability of being loss-of-function intolerant (pLI) scores are used for the assessment of genetic pathogenicity (PMID: 30962618), and inconsistencies between pLI scores and pathogenic gene findings are mentioned in the Alban et al. manuscript (PMID: 30977936), e.g. DISC1 for the schizophrenia pathogenic gene, deafness gene TECTA, the epileptic encephalopathy causative gene CNTNAP2 and the intellectual disability gene PPM1D all had pLI scores close to 0; Alban et al. mentioned that pLI scores should be used with caution when interpreting missense variants. We downloaded the variants of the CTTNBP2 gene included in the gnomAD database (see Supplementary Table 8) and performed the statistics, and we found that in the East Asian population Allele Frequency of missense mutations in the CTTNBP2 gene ranged from 0 to 0.0019 , which is relatively rare; notably, we found that Allele Frequency of missense mutations in the CTTNBP2 gene located in the coiled-coil (CC) domain ranged from 0.000283 to 0.00000397 , which is very rare. The coiled-coil domain mediates the oligomerization of CTTNBP2 and regulate spine formation. We made the corresponding modifications.

- The DOMINO score is of 0.096 (probability to be associated with a dominant disorder)

Reply: Thank you for your comment. We know that the genetic model of autism is very complex, and the genes identified so far in autism sometimes fit the autosomal recessive inheritance pattern, some times fit the autosomal dominant inheritance pattern, and some times are difficult to be explained by the conventional inheritance pattern. The de novo mutation M115T in CTTNBP2 gene we found in the proband may explain the autism phenotype, and considering that the DOMINO score is of 0.096, it suggests that the probability that the genetic model of this de novo mutation is likely to be an autosomal dominant pattern is very low. We have revised the manuscript accordingly.

- There is 39 variants in SFARI database but only 3 of them are de novo and only one of them is a missense variant.

Reply: Thanks for the comment, indeed the percentage of de novo mutations among all CTTNBP2 mutations included in SFARI database is very low, suggesting that de novo mutations are relatively rare.

- Finding a de novo variant does not imply pathogenicity. Every individual carry 1 to 3 coding de novo variant and most of them are benign. According to ACMG guidelines, the variant can only be classified as VUS (PS2, PM2, PP3).

Reply: I agree with the reviewer comment that the discovery of a de novo mutation does not necessarily imply pathogenicity, considering that each individual carries 1-3 de novo mutations and most of them are benign; with reference to the reviewer's suggestion we have made the corresponding changes.

- The variant M120I identified by Shih et al. was not found de novo but maternally transmitted (in SFARI).

Reply: Thanks for the comment, Shih et al. manuscript (PMID: 35562389) mentions that variant M120I is a heterozygous mutation and the authors introduced the heterozygous mutation into mice and observed that the mutation impaired the social behavior of the mice.

In conclusion without either experimental data proving a gain-of-function mechanism of the missense variant or other cases with missense de novo variant in this gene, there is not enough evidence to link the variant to the phenotype. Otherwise, we could sequence any individual, find de novo variant with a link to a disease and claim that there is an causal effect.

Reply: Thanks for the comments, In the present study, we completed whole-exome sequencing in a quad family affected by autism, and by systematic bioinformatics analysis, we identified a de novo mutation p.M115T in the CTTNBP2 gene; we speculate that this variant may be linked to autism, mainly considering that: 1) the CTTNBP2 gene is a strong autistic candidate, meanwhile de novo mutation p.M115T is relatively rare in normal population, including the fact that low frequency of CTTNBP2 gene missense mutation in gnomAD East Asians population, and that p.M115T was not detected in 867 normal Chinese; 2) the de novo mutation p.M115T is located in the Coin-Coin region of the CTTNBP2 gene, which has been previously reported gene adjacent to the M120I variant introduced into mice resulted in impaired social behavior in mice. Indeed, finding de novo variants associated with the disease and claiming a causal effect will require further functional validation. This is a limitation of the manuscript, and we have modified the discussion accordingly.

Reviewer 2 Report

The authors describe a nuclear family with a single child affected with an autism spectrum disorder.  The authors detected a single de novo variant in the CTTNBP2 gene and suggest that this variant is responsible for the phenotype seen in the patient.

There are several points that must be clarified by the authors prior to the publication of this manuscript.

1) The manuscript is written in a manner that suggests that the authors were only looking for variants that might affect dendritic spines and thus identified the CTTNBP2 as causal.  A more complete explanation of the WES results and how the authors arrived at the conclusion that the CTTNBP2 variant is responsible for the clinical features seen in the proband must be included.  It is unlikely that the CTTNBP2 variant was the only de novo variant observed in the proband, but the supplementary data do not give a clear description of the complete set of whole exome data. 

2) Though the identified variant is rare based on a small set of 867 control individuals genotyped, larger public databases such as the gnomAD database should be used to better understand the presence or absence of this variant (and other variants at the same position) in the general population.  For example, gnomAD reveals the presence of a p.M115L variant in a single individual based on whole genome sequencing, suggesting that variation at this amino acid may not be pathogenic, at least in some individuals.

3) The analysis process suggests that the authors based their data evaluation on the SFARI database, containing genes that already have been linked to autism spectrum disorders, but they also stated that not all autism genes have yet been discovered.  Thus the authors need to include a more full description of potential candidate causal variants in genes not found in the SFARI database but potentially involved in pathways that have been linked to autism to support their conclusion that the p.M115T variant they observed is the causal variant in the proband.

4) The authors do not provide any strong evidence for the pathogenicity of the p.M115T variant.  The data cited are suggestive, but do not provide any conclusive evidence.  This the authors should not over-state the significance of this finding.

5) At least a brief discussion of the genetic models evaluated for the patient should be included.  Did the authors only look for de novo variants or were autosomal recessive variants considered as well?  Were common variant models evaluated? Much genetic risk has been attributed to common genetic variants working collectively resulting in autism have been suggested.  Many of these points could be addressed by including a complete description of the results in the supplemental materials and a further discussion of the genetics of autism in the main manuscript.

6) A minor point is that many of the authors' sentences are too long and should be divided into two or more sentences.  This would help make the manuscript easier to read.  Examples are highlighted in the PDF with comments indicating the need for better writing.  Also on page 2 line 49, the word "nearly" which is highlighted in the PDF, should be deleted. On page 2 line 84, dendritic spines are not substrates for the pathogenesis of autism, but their alteration may be involved in the pathogenesis of ASD.  On page 3 line 131, the term should be milliliters, not microliters, and similarly on page 4 line 152, the authors mean micrograms, not microliters. Page 4 line 165 should say" identify the de novo and rare..." instead of "determine the de novo and rare ...".  On page 4 line 167, variants with a MAF >0.05 were removed, not <0.05.

Author Response

The authors describe a nuclear family with a single child affected with an autism spectrum disorder.  The authors detected a single de novo variant in the CTTNBP2 gene and suggest that this variant is responsible for the phenotype seen in the patient.

There are several points that must be clarified by the authors prior to the publication of this manuscript.

1) The manuscript is written in a manner that suggests that the authors were only looking for variants that might affect dendritic spines and thus identified the CTTNBP2 as causal.  A more complete explanation of the WES results and how the authors arrived at the conclusion that the CTTNBP2 variant is responsible for the clinical features seen in the proband must be included.  It is unlikely that the CTTNBP2 variant was the only de novo variant observed in the proband, but the supplementary data do not give a clear description of the complete set of whole exome data. 

Reply: Thank you for pointing out this issue. In order to find genetic mutations associated with autism, we first examined the genome-wide copy number variation (CNV) by chromosome microarray (CMA) technology; for patients with normal results or CNV of uncertain significance (VOUS), further whole-exome sequencing (WES) of the family was performed. In this study, CMA did not find clinically significant copy number variants in the proband (Supplementary Figure 1), and the average sequencing depth of the WES achieved samples was 151.1975 (range 116.94 -171.45 ), with 98.65% of the target regions achieving more than 20-fold coverage, as shown in Supplementary Table 2 for the results; by systematic WES bioinformatics analysis, 22218.5 (range 21957- 22401) single nucleotide variants and 690 (range 664-716) short insertion deletion variants were found in each sample, and the results are shown in Supplementary Table 3 and Supplementary Table 4. Considering that we have whole-exome sequencing data for the complete quad family, we can easily distinguish the variants found in the proband into inherited or de novo variants, and all the variants in the proband were used together to find clinically relevant variants in autism, and the general process was as follows: first, we filtered to common variants with an allele frequency greater than 0.05; then, we evaluated the pathogenicity of the variants with reference to the ACMG guidelines, together with the SFARI gene score to identify candidate clinically relevant variants for autism; Meanwhile, considering the limited data of Chinese population within East Asians in gnomAD databases, we validated clinically relevant variants that could be candidates for autism in 867 normal populations; of course, we also combined the findings of Alphafold for structural prediction of the variants, and Shih et al. (PMID: 33168105). In conclusion, we hypothesized that the de novo mutation p.M115T in the CTTNBP2 gene identified in the proband might explain its phenotype, considering that CTTNBP2 is a known candidate gene for strong autism. We have revised the manuscript accordingly.

2) Though the identified variant is rare based on a small set of 867 control individuals genotyped, larger public databases such as the gnomAD database should be used to better understand the presence or absence of this variant (and other variants at the same position) in the general population.  For example, gnomAD reveals the presence of a p.M115L variant in a single individual based on whole genome sequencing, suggesting that variation at this amino acid may not be pathogenic, at least in some individuals.

Reply: Thanks for the comment, this is a great suggestion. We downloaded the CTTNBP2 gene variants included in the gnomAD database (see Supplementary Table 8) and performed the statistics. We found that the allele frequency of missense mutations in the CTTNBP2 gene ranged from 0 to 0.0019 in the East Asian population, which is relatively rare; notably, we found that the allele frequency of missense mutations in the CTTNBP2 gene located in the coiled-coil (CC) region ranged from 0.000283 to 0.00000397, which is very rare. Notably, we found the presence of variant p.Met115Lys in the CTTNBP2 gene, which is rare in the population (only 1 detected in 31392 individuals) and rare in the East Asian population, (only 0 detected in 1560 East Asians). Considering that the collection of Chinese population may be limited in the East Asian population of the gnomAD database, we performed further large-scale valida-tion of M115T in the Chinese population, and the MassARRAY also confirmed that the variant was rare (0 of 867) in Chinese individuals not affected by ASDs (Supplementary Table 9)

3) The analysis process suggests that the authors based their data evaluation on the SFARI database, containing genes that already have been linked to autism spectrum disorders, but they also stated that not all autism genes have yet been discovered.  Thus the authors need to include a more full description of potential candidate causal variants in genes not found in the SFARI database but potentially involved in pathways that have been linked to autism to support their conclusion that the p.M115T variant they observed is the causal variant in the proband.

Hyperinsulinism and Byzanthine arch palate pathways. We have revised the manuscript accordingly。

Reply: After initially filtering the public database for common variants and synonymous variants, a total of 541 variants were obtained (see Supplementary Table 5) from 248 genes, and after removing the SFARI genes (Supplementary Table 6, total 1095 genes), a total of 218 genes were obtained, and we performed pathway enrichment analysis for these genes using Metascape (PMID: 30944313), and the results are shown in Supplementary Figure 2. It can be seen that all 248 genes were enriched to Delayed speech and language development and Byzanthine arch palate pathways, and the 218 genes after removal of SFARI genes were enriched to Hyperinsulinism and Byzanthine arch palate pathways. We have revised the manuscript accordingly.

4) The authors do not provide any strong evidence for the pathogenicity of the p.M115T variant.  The data cited are suggestive, but do not provide any conclusive evidence.  This the authors should not over-state the significance of this finding.

Reply: Thanks for the comment, I agree with the reviewer, considering that the CTTNBP2 gene is a known strong autism candidate gene in SFARI database, we identified a rare de novo missense mutation p.M115T in the CTTNBP2 gene in patients affected by autism, which is located in the N-terminal coiled-coil (CC) region, which mediates the oligomerization of the CTTNBP2 protein and thus regulates spine formation, and the next step In-depth functional validation is required to confirm whether this variant directly affects spine formation and the causal relationship with autism. We have revised accordingly.

5) At least a brief discussion of the genetic models evaluated for the patient should be included.  Did the authors only look for de novo variants or were autosomal recessive variants considered as well?  Were common variant models evaluated? Much genetic risk has been attributed to common genetic variants working collectively resulting in autism have been suggested.  Many of these points could be addressed by including a complete description of the results in the supplemental materials and a further discussion of the genetics of autism in the main manuscript.

Reply: Thanks for the comment. Considering the complexity of the genetic model of autism, the currently identified autism genes sometimes conform to an autosomal recessive genetic model, some times to an autosomal dominant genetic model, and some times are difficult to explain by the conventional genetic model. The prediction of whether the CTTNBP2 gene conforms to a dominant inheritance pattern was performed by DOMINO software (PMID: 28985496), and the Probability of being autosomal dominant (AD) score is 0.096, suggesting that the inheritance pattern of this new mutation may be "Very likely recessive". Common genetic variants do play an important role in autism, and considering the limited sample of this study, we did not focus on common genetic variants, which is a limitation of this study. We have revised accordingly with reference to the reviewers' comments.

6) A minor point is that many of the authors' sentences are too long and should be divided into two or more sentences.  This would help make the manuscript easier to read.  Examples are highlighted in the PDF with comments indicating the need for better writing.  Also on page 2 line 49, the word "nearly" which is highlighted in the PDF, should be deleted. On page 2 line 84, dendritic spines are not substrates for the pathogenesis of autism, but their alteration may be involved in the pathogenesis of ASD.  On page 3 line 131, the term should be milliliters, not microliters, and similarly on page 4 line 152, the authors mean micrograms, not microliters. Page 4 line 165 should say" identify the de novo and rare..." instead of "determine the de novo and rare ...".  On page 4 line 167, variants with a MAF >0.05 were removed, not <0.05.

Reply: Thanks for pointing out this issue, I don't see the PDF file marked up, and my understanding is that the reviewer has specified all the issues here, and we have revised it accordingly.

Round 2

Reviewer 1 Report

The authors answered my comments and changed the conclusions.

Author Response

Thank you for your constructive comments on this study.

Reviewer 2 Report

The authors have made significant improvements to the manuscript, and clearly demonstrated the possibility that CTTNBP2 is a candidate gene for ASD.  

The authors now have used DOMINO to predict that mutations affecting CTTNBP2 are likely to function in an autosomal recessive manner.  Since no second mutation was detected in the gene, it is unclear how the variant they detected explains the clinical features in the patient. It is possible that a dominant interfering mechanism could be relevant here, but this molecular model was not discussed, nor were other potential variants in the gene identified, so an autosomal recessive model is inconsistent with their findings being relevant to this patient.  

Other specific comments:

1) Abstract line 29, "families" should be "family", since only 1 family was evaluated.

2) Abstract line 32, variant confirmation by MassArray and Sanger sequencing ca  be deleted from the abstract since it does nothing to validate the clinical interest of the findings.

3) The sentence started on Abstract line 36 whould end with "... spine formation." on line 37, and the remainder of the sentence should be split into 2 additional sentences for clarity.

4) Abstract line 40 should be modified to say "candidate clinically relevant variant" as there is no direct evidence presented for the clinical relevance of the variant discussed.

5) Introduction lines 58-66 should be split into multiple sentences.

6) The final sentence of the introduction, starting on line 94, should be split into 2 sentences.  A single sentence, as currently written, should not be place into a separate paragraph, but splitting this into two sentences clarifies the text and creates a valid paragraph.

7) Introduction line 137 should say "... and actin cytoskeletal proteins" instead of "... or actin cytoskeletal proteins".

8) Methods line 180 should be "regions of homozygosity" rather than "regions of homogeneity"

9) Methods line 186 should be ended after "CNV database".  Since no interesting genes were found the remainder of the sentence can be deleted.

10) Methods Line 230-231 first sentence should end after "frequency".  The second sentence should start "The Magibind DNA ... was used tjo isolate DNA from these individuals."  Third sentence should be "The CTTNBP2 variant was evaluated using the MassARRAY ...".

11) The genetic model prediction section, lines 240-243, should be split into multiple sentences and stated more clearly.  As written, the section is difficult to interpret.

12) The relevance of section 2.9 is unclear since only a single gene is being discussed in the manuscript.  What genes are being analyzed?

13) Results line 255 should say "... were subjeted to..." rather than "... was subjected to...".

14) As written, lines 258-260 suggest each individual sequenced had exactly the same number of variants.  This clearly is not the case and needs to be clarified.

15) Line 261 should read "After initially filtering the results to remove common and synonymous variants, 541 variants remained."  In the remainder of this section, it is not clear why the SFARI genes were removed.  Since these genes are ASD candidate genes, were the authors only focused on the CTTNBP2 variant?  If no likely pathogenic or pathogenic variants were observed in the removed genes, that should be stated as the reason, rather than stating that they were remopved because they were found in the SFARI database.  This section should be re-written so that the results and meaning are more clear.  In addition, on line 274, variants in other ASD candidate genes are mentioned, but none of them are described.  Is there a reason these were not discussed in the results section?

16) The first paragraph of the discussion sections seems to state that the authors were involved in the work cited in references 11 and 61.  This paragraph is unclear and should be worded more carefully to better describe their meaning.

17) Discussion lines 376-384 should be split into multiple sentences to facilitate easier reading and comprehension.

18) The following paragraph discussing the results using DOMINO suggest an autosomal recessive model for loss of CTTNBP2, but no variant is identified in the second copy of the gene, so either a different model is involved or the variant discussed is not the relevant variant in this patient.  This section needs to be clarified.

19) "Dendritic spines located at the locations of ..." on line 404 should be shortened to remove the redundant use of "location".

20) NF1 encodes neurofibromin, not neurofibrin, on line 410.

Author Response

See attached PDF file. 

Round 3

Reviewer 2 Report

The authors have made significant improvements to the manuscript.  The manuscript is now much easier to read and the points the authors make with respect to genetic models and autism are valid.

One minor point with respect to the discussed autosomal dominant model for the CTTNBP2 variant detected is that the variant could be functioning molecularly as recessive while functioning genetically as dominant.  For example, if the detected variant is causing excessive polymerization, of CTTNBP2, there may be little oligomeric or monomeric CTTNBP2 available to interact with other cellular interactors, resulting in insufficient CTTNBP2 functional activity.  This is molecularly recessive, consistent with the results of the DOMINO analysis, but also would be genetically dominant.